# LMGQS: A Large-scale Dataset for Query-focused Summarization

**Ruochen Xu, Song Wang, Yang Liu, Shuohang Wang, Yichong Xu, Dan Iter**

**Pengcheng He, Chenguang Zhu, Michael Zeng**
Microsoft Azure AI
{ruox,sonwang,yaliu10,shuowa,yicxu,iterdan}
{penhe,chezhu,nzeng}@microsoft.com

## Abstract

Query-focused summarization (QFS) aims to extract or generate a summary of an input document that directly answers or is relevant to a given query. The lack of large-scale datasets in the form of documents, queries, and summaries has hindered model development in this area. In contrast, multiple large-scale high-quality datasets for generic summarization exist. We hypothesize that there is a hidden query for each summary sentence in a generic summarization annotation, and we utilize a large-scale pretrained language model to recover it. In this way, we convert four generic summarization benchmarks into a new QFS benchmark dataset, LMGQS, which consists of over 1 million document-query-summary samples. We thoroughly investigate the properties of our proposed dataset and establish baselines with state-of-the-art summarization models. By fine-tuning a language model on LMGQS, we achieve state-of-the-art zero-shot and supervised performance on multiple existing QFS benchmarks, demonstrating the high quality and diversity of LMGQS.

## 1 Introduction

The field of generic summarization (See et al., 2017; Gehrmann et al., 2018; Liu and Lapata, 2019) has made significant progress in recent years, thanks to the development of generative deep neural models (Sutskever et al., 2014; Vaswani et al., 2017) and the availability of large-scale training data (Nallapati et al., 2016; Narayan et al., 2018; Zhu et al., 2021). However, query-focused summarization (QFS) presents a significant challenge due to the lack of data. Most of the available QFS corpora (Dang, 2006a,b; Nema et al., 2017; Baumel et al., 2016; Zhong et al., 2021) contain only a few thousand documents or less, which is insufficient for training a robust neural model.

We propose a **L**anguage **M**odel **G**enerated **Q**uery-focused **S**ummarization Dataset (LMGQS)

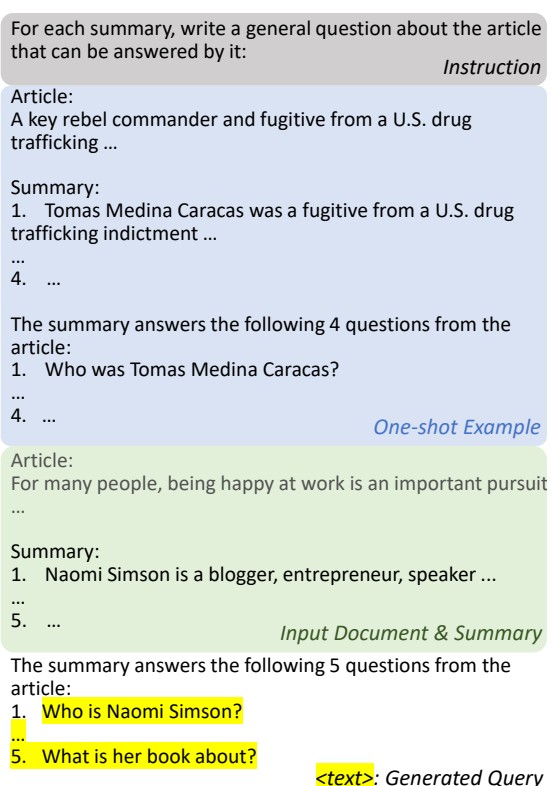

For each summary, write a general question about the article that can be answered by it:
*Instruction*

Article:
A key rebel commander and fugitive from a U.S. drug trafficking ...

Summary:
1. Tomas Medina Caracas was a fugitive from a U.S. drug trafficking indictment ...
...
4. ...

The summary answers the following 4 questions from the article:
1. Who was Tomas Medina Caracas?
...
4. ...
*One-shot Example*

Article:
For many people, being happy at work is an important pursuit ...

Summary:
1. Naomi Simson is a blogger, entrepreneur, speaker ...
...
5. ...
*Input Document & Summary*

The summary answers the following 5 questions from the article:
1. Who is Naomi Simson?
...
5. What is her book about?
*<text>: Generated Query*

Figure 1: Example prompt for query generation. The top part is the instruction, followed by the one-shot example consisting of document, summary, and query. The query for the input document (highlighted in yellow) and summary is generated by InstructGPT.

to address the lack of a large-scale QFS dataset. Human annotation for QFS typically involves generating suitable queries and then writing corresponding summaries, which is both time-consuming and expensive. Furthermore, it may necessitate a meticulous definition of the query scheme based on the domain of documents (Zhong et al., 2021). We hypothesize that, for a pair of document and summary in a generic summarization dataset, hidden queries exist that represent the information needs associated with the summary. Therefore, to efficiently scale up annotation, we prompt the large-scale language model InstructGPT (Ouyang et al.,

| Prompt Query Type | Dataset | do does did | is/are was were | can could | will would | have has had | what | when | where | who whom | which | whose | why | how | other | Yes/no queries | Wh-queries |
|---|---|---|---|---|---|---|---|---|---|---|---|---|---|---|---|---|---|
| Wh-queries | CNN/DM | 0.08 | 0.16 | 0.01 | 0.02 | 0.02 | 52.36 | 5.36 | 4.29 | 14.21 | 0.2 | 0 | 3.98 | 19.21 | 0.1 | 0.29 | 99.61 |
| | XSUM | 0.06 | 0.09 | 0.01 | 0.09 | 0.01 | 72.54 | 2.07 | 0.98 | 9.66 | 0.16 | 0 | 2.15 | 12.01 | 0.17 | 0.26 | 99.57 |
| | SAMSum | 0.49 | 0.19 | 0.03 | 0.12 | 0.04 | 66.43 | 7.12 | 6.58 | 7.6 | 0.12 | 0 | 2.79 | 8.29 | 0.19 | 0.87 | 98.93 |
| | DialogSum | 0.44 | 0.13 | 0.04 | 0.02 | 0.02 | 78.88 | 1.4 | 2.82 | 3.8 | 0.12 | 0 | 3.88 | 8.35 | 0.08 | 0.65 | 99.25 |
| Yes/no queries | CNN/DM | 40.33 | 43.06 | 1.54 | 5.79 | 4.42 | 0.12 | 0.17 | 0.02 | 0.15 | 0.01 | 0 | 0.01 | 0.08 | 4.32 | 95.14 | 0.56 |
| | XSUM | 26.56 | 42.19 | 1.41 | 12.2 | 8.07 | 0.17 | 0.09 | 0.01 | 0.13 | 0 | 0 | 0 | 0.05 | 9.12 | 90.43 | 0.45 |
| | SAMSum | 37.96 | 36.78 | 0.79 | 17.88 | 2.52 | 0.03 | 0.03 | 0.01 | 0 | 0 | 0 | 0 | 0.01 | 3.99 | 95.93 | 0.08 |
| | DialogSum | 59.68 | 27.84 | 0.56 | 4.9 | 1.11 | 0.01 | 0 | 0 | 0 | 0 | 0 | 0 | 0 | 5.9 | 94.09 | 0.01 |

Table 1: Breakdown of query types on QuerySum. The upper part of the table shows the query type percentage when using wh-queries in the one-shot prompt example. The lower part shows the percentage when prompting with yes/no queries. Color of blue means a higher percentage while red means a lower one.

2022) with documents and summaries from four generic summarization datasets to generate the hidden queries. This approach results in the LMGQS dataset, which contains over 1.1 million triplets of document, query, and summary, encompassing a wide range of document and question types.

To investigate the utility of our proposed LMGQS , we finetune a pretrained language model on it. The model accepts the concatenation of the original document and generated query as input and is trained to produce the original summary. We then compare the finetuned model with various query-focused summarization models on several existing QFS benchmarks that have no overlap with LMGQS under the zero-shot setting. Empirical results demonstrate that the model finetuned on LMGQS achieves promising performance on both single-document and multi-document QFS benchmarks, surpassing strong baselines. Similarly, when utilizing LMGQS for pre-finetuning, the model achieves state-of-the-art performance in the supervised setting.

In summary, our contributions are three-fold: (1) We introduce a novel framework for constructing a QFS dataset by converting existing generic summarization datasets using language models as annotators. (2) We present LMGQS, a large-scale QFS benchmark, to foster future research on QFS[1]. (3) The model finetuned on LMGQS exhibits robust generalization capability and achieves remarkable zero-shot and supervised performance on other unseen QFS test sets.

## 2 Dataset Creation

We choose 4 generic datasets to build LMGQS: CNN/DailyMail (Nallapati et al., 2016), XSUM (Narayan et al., 2018), SAMSum (Gliwa et al., 2019), and DialogSum (Chen et al., 2021). Among

them, CNN/DailyMail and XSUM are news summarization datasets, where both the documents and summaries are in formal written English. SAMSum and DialogSum are two recently proposed dialogue summarization datasets, whose inputs are the transcripts of multi-speaker conversations.

### 2.1 Prompt-based Query Generation

Given a document and its corresponding summary, we take advantage of the robust few-shot capabilities of InstructGPT (Ouyang et al., 2022) to generate a query that encapsulates the information required by the annotator when crafting the summary. More specifically, we construct a prompt for each document-summary pair and input it into the InstructGPT model to generate the query by completing the prompt. An example prompt is illustrated in Figure 1. Since InstructGPT excels at adhering to human-readable instructions and can even generalize to unseen instructions (Ouyang et al., 2022), we begin our prompt with a clear directive for the query generation task. Following the instruction, we incorporate a one-shot example of the task into the prompt, which includes a human-written query derived from a document-summary pair. We set the number of examples to be 1 based on a balance between effectiveness and efficiency: during our preliminary exploration, we noticed more failure cases for zero-shot query generation, while incorporating additional examples in the prompt would increase both the time and cost of generation.

In the one-shot example, we restrict the number of queries to be equivalent to the number of summaries. In other words, there exists a one-to-one correspondence between the sentences in the summary and the query. This constraint is imposed by appending prefix indices and appending newline characters for each summary/query sentence, as illustrated in Figure 1.

Due to the domain difference between news and

[1]Dataset will be released after the anonymous period

| Prompt Query Type | Dataset | Count | Len (doc) | Len (query) | Len (sum) | NTP (sum, doc) | NTP (query, doc) | NTP (doc, sum) | NTP (doc, query) | NTP (query, sum) | NTP (sum, query) |
|---|---|---|---|---|---|---|---|---|---|---|---|
| Wh-queries | CNN/DM | 311938 | 677.7 | 29.9 | 52.0 | 21.5 | 36.7 | 87.34 | 94.74 | 44.6 | 70.3 |
| | XSUM | 226287 | 368.0 | 10.0 | 22.1 | 42.2 | 43.1 | 92.37 | 96.45 | 45.9 | 73.4 |
| | SAMSum | 16368 | 92.9 | 16.2 | 22.3 | 41.5 | 56.5 | 76.51 | 87.89 | 43.5 | 59.2 |
| | DialogSum | 14460 | 130.7 | 15.1 | 24.2 | 36.6 | 48.0 | 82.7 | 90.97 | 47.0 | 65.9 |
| Yes/no queries | CNN/DM | 311920 | 677.7 | 33.7 | 52.0 | 21.5 | 27.4 | 87.34 | 92.6 | 20.7 | 48.2 |
| | XSUM | 226276 | 368.0 | 10.7 | 22.1 | 42.2 | 40.4 | 92.37 | 95.91 | 21.4 | 57.8 |
| | SAMSum | 16368 | 92.9 | 17.5 | 22.3 | 41.5 | 42.9 | 76.51 | 81.71 | 19.0 | 34.1 |
| | DialogSum | 14460 | 130.7 | 16.4 | 24.2 | 36.6 | 37.9 | 82.7 | 87.91 | 27.8 | 47.9 |

Table 2: Statistics about LMGQS. "Len" stands for length and Len(string) is the count of word tokens in the string. "NTP" stands for novel token percentage. NTP(string1, string2) computes the percentage of tokens in string1 that are not present in string2.

dialogue summarization, we choose different one-shot examples for the two domains. The queries for the two example pairs were annotated by the authors of this paper and are attached in the appendix.

## 2.2 Prompt Query Types

Given a document and a summary sentence, multiple valid queries can be formulated. For instance, consider the summary sentence: *She has released a book to encourage people to find their passion at work.* One possible query is: *What is her book about?* Alternatively, another valid query could be: *Has she released a book?* To address this variety, we utilize two sets of annotated queries: yes/no queries and wh-queries. Yes/no queries correspond to questions that can be answered with a simple "yes" or "no". However, in the context of QFS, the summary (i.e., the answer to the yes/no query) is never a mere "yes" or "no". For example, for a yes/no query like *Is he still alive?*, we expect the answer to be: *He was killed in an attack on a guerrilla encampment* rather than a simple *no*. Detailed annotated queries are presented in Table 11.

The type of queries in a one-shot prompt significantly influences the generated queries. We provide a breakdown of query types in Table 1. It is evident that when the prompt includes only wh-queries, over 99% of the generated queries are also wh-queries, with the most frequent ones beginning with "What". The same pattern applies when the prompt contains only yes/no queries. The most common queries generated by InstructGPT typically start with "do/does/did" or "is/are/was/were".

## 2.3 Statistics of LMGQS

Using the aforementioned prompting method, we collected 1,138,077 document-query-summary triples covering 13 different query types. Detailed statistics of the generated LMGQS dataset are shown in Table 2. First, the length of the generated

queries has a strong Pearson correlation (0.95) with the length of summaries, which is expected due to our one-to-one mapping between the summary and query sentences. Second, the length of queries is consistently shorter than the summary, with wh-queries slightly shorter than yes/no queries.

We introduce the novel token percentage: $NTP(string1, string2)$, defined as the percentage of tokens in string1 that are absent in string2. This statistic quantifies the amount of unique information contained in string1 with respect to string2 First, $NTP(doc, query)$ is always lower than $NTP(doc, sum)$, indicating that the generated query always contains less information about the document than the summary. Subsequently, we observe that $NTP(query, doc)$ is in general higher than $NTP(sum, doc)$, because queries are shorter and contain more unique question words like "what" and "did". Finally, $NTP(query, sum)$ being considerably lower than $NTP(sum, query)$ shows that the summary contains more unique information than the query. Furthermore, the query includes a subset of information present in the summary. For instance, a query might inquire about a specific entity in the document, while the summary addresses the query with detailed contexts and facts extracted from the document.

In conclusion, LMGQS encompasses documents in both written and spoken languages, covering a wide range of document/summary lengths, abstraction levels, and compression ratios.

## 3 LMGQS for QFS

In this section, we demonstrate that by finetuning pretrained language models on LMGQS, one can obtain a QFS model that generalizes effectively to unseen tasks and domains. In particular, we finetuned a BART model (Lewis et al., 2020), and the resulting model, LMGQS BART, exhibits promis-

| Dataset | Size(Train/Test) | Query Example |
|---|---|---|
| MultiOpEd | 1954/560 | is protecting the environment incompatible with capitalism's values? |
| NEWTS - word | 2400/600 | snow, weather, cold, winter, temperatures, conditions, hot, morning, expected, parts |
| NEWTS - phrase | 2400/600 | winter temperatures, hot weather conditions, a cold morning, snow is expected later |
| NEWTS - sentence | 2400/600 | This topic is about winter temperatures as opposed to hot weather conditions, cold mornings, and weather forecasts like snow being expected later. |
| Debatepedia | 12000/1000 | Would the election of a president make the eu a more accountable institution? |
| DUC 2006 | 0/200 | Identify computer viruses detected worldwide. Include such details as how they are spread, what operating systems they affect, what damage they inflict, their country of origin, and their creators wherever possible. |
| DUC 2007 | 0/180 | Describe the state of teaching art and music in public schools around the world. Indicate problems, progress and failures. |

Table 3: Size and example queries of QFS datasets used in evaluation.

ing performance on various QFS datasets when directly applied to the unseen test set. Moreover, when extending the fine-tuning process with several thousand in-domain QFS data points, the resulting supervised model surpasses other strong supervised baselines.

## 3.1 Implementation Details

We fine-tuned BART-Large (Lewis et al., 2020) on LMGQS , using a maximum input length of 1024 and output length of 256. The input string consists of a document and a query, formatted as *question:\n <query> \n context:\n<document>*, where "\n" represents a newline character. We employed 8 NVIDIA Tesla V100 GPUs for training, with a batch size of 4 per GPU and an accumulation step of 8, yielding an effective batch size of 256. The BART model was fine-tuned using a learning rate of $3 \times 10^{-5}$ for $50,000$ steps, and the learning rate was scheduled by a polynomial scheduler with 2000 warmup steps. We set a weight decay of $0.001$ and a label smoothing factor of $0.1$. For supervised finetuning, we continued to finetune the LMGQS BART model with 2000 total steps and 200 warm-up steps. The implementation from Huggingface (Wolf et al., 2020) was utilized.

## 3.2 Datasets

We conduct evaluation of the finetuned BART-Large model (LMGQS BART) on several existing QFS benchmark datasets.

- MultiOpEd (Liu et al., 2021) presents an open-domain news editorial dataset specifically designed to support automatic perspective discovery in news articles. Given a query that ex-

plicitly addresses a controversial topic, a system is expected to generate a single-sentence thesis statement that summarizes the arguments presented. Along with ROUGE scores as evaluation metrics, the paper also proposes trained classifiers to assess the correctness and relevance of the generated summary. More specifically, a stance classifier is utilized to predict whether a summary shares the same stance as the news article. For example, a summary that presents an opposing argument to the article might still achieve a high ROUGE score due to n-gram overlap but would receive a low stance accuracy. Similarly, a relevance classifier is employed to evaluate whether the summarized perspective is pertinent to the query.

- NEWTS (Bahrainian et al., 2022) dataset is a corpus for summarizing news topics. It is based on the CNN/Dailymail dataset (See et al., 2017) and was annotated through online crowd-sourcing. Each source article is paired with two reference summaries, each focusing on a different theme of the source document. The dataset has 3,000 source articles (2,400 for training, and 600 for testing). In addition to standard ROUGE score, the dataset is evaluated using a LDA topic model indicating the strength of the target topic for the generated summary. We follow the implementation from Bahrainian et al. (2022) to compute the topic focus score. It is expected that summaries closer to the target topic get higher topic focus scores.

| | Model | ROUGE1 | ROUGE2 | ROUGEL | BERTSCORE | Relevance. Acc. | Stance Acc. |
|---|---|---|---|---|---|---|---|
| Supervised Models | BART | 28.2 | 11.3 | 27.0 | 88.7 | 91.9 | 72.3 |
| | + Rel | 28.4 | 11.5 | 27.1 | 88.7 | 93.0 | 72.7 |
| | + Stance | 28.2 | 11.5 | 26.9 | 88.8 | 91.3 | 73.4 |
| | + Rel & Stance | 29.2 | 11.9 | 27.9 | 88.7 | 94.6 | 74.3 |
| | CNN/DM BART | 26.9 | 10.7 | 25.2 | 86.4 | **99.5** | 68.6 |
| | LMGQS BART | **31.5** | **13.8** | **29.8** | **89.1** | 96.8 | **78.8** |
| Zero-shot Models | InstructGPT 002 | 23.8 | 9.3 | 21.9 | 85.7 | **99.5** | 73.2 |
| | CNN/DM BART | 17.9 | 5.2 | 16.2 | 85.2 | 94.8 | 62.9 |
| | LMGQS BART | **25.4** | **10.4** | **23.6** | **87.2** | **99.5** | **77.0** |

Table 4: ROUGE scores and accuracies of stance and relevance on MultiOpEd dataset. All baseline results except for InstructGPT are from (Liu et al., 2021)

- Debatepeida (Nema et al., 2017) was built on Debatepedia - an encyclopedia of pro and con arguments and quotes on critical debate topics. The summaries are highly abstractive and not extractive in the sense that the summary does not necessarily comprise of a sentence which is simply copied or shortened from the original document.

- Document Understanding Conferences (DUC) 2006/2007 [2] set up the task to simulate real-world complex question answering. The query in this dataset cannot be answered by simply stating a name, date, quantity, etc. Given a topic and a set of 25 relevant documents, the task is to synthesize a fluent, well-organized 250-word summary of the documents that answers the question(s) in the topic statement.

### 3.3 Baselines

We compare LMGQS BART with the following baseline models:

CNN/DM BART is a large BART model that has been fine-tuned on the query-agnostic CNN/DailyMail dataset (See et al., 2017). This serves as a baseline model for summarization based solely on the input document, without considering the query in the QFS setting.

InstructGPT 002 is an InstructGPT model that can be accessed through the OpenAI API using the text-davinci-002 model. A simple template, "Summarize by answering the following questions:", is used to link the document with the query and generate content with the temperature set to 1.0, top-p set to 0.9, and maximum length set to 512.

LaQSUM (Xu and Lapata, 2022) is a recent model that learns latent queries from documents

for abstractive summarization. Unlike other approaches, LaQSUM models the query as hidden binary variables to indicate whether a token in the document contributes to the information sought in the summary. This model does not require QFS annotation and is trained on the CNN/DM dataset.

MARGESUM (Xu and Lapata, 2021) is a state-of-the-art few-shot method for QFS that requires a small QFS development set.

GSUM+Query is adapted from GSUM (Dou et al., 2021), which is a guided summarization system. An unsupervised query-focused extractive system is used to pre-extract the top-ranked sentences for each test document as guidance. The GSUM model is trained with the CNN/DM dataset.

QuerySum (Xu and Lapata, 2020) is an extractive method that uses QA datasets as distant supervision to train an evidence estimator for identifying segments likely to answer the query and should be included in the summary.

ProphetNet (Qi et al., 2020) is a supervised abstractive summarization model that predicts the next n tokens simultaneously. The results for ProphetNet are taken from the NEWTS paper (Bahrainian et al., 2022).

Unsupervised extractive baselines are taken from Xu and Lapata (2022). Lead and LexRank estimate sentence-level centrality using Markov Random Walk on graphs.

QMDSCNN (Pasunuru et al., 2021) transfers the CNN/DailyMail dataset into query-focused multi-document summarization dataset and build abstractive end-to-end neural network models to obtain zero-shot results on DUC 2016 and 2017 datasets.

### 3.4 Query Unification

Different QFS datasets have different query formats. For instance, Debatepedia has the query format of a natural question, which is the same as

[2]URL at https://www-nlpir.nist.gov/projects/duc

LMGQS, while the majority of queries in DUC datasets are instructions such as "*Discuss conditions on American Indian reservations or among Native American communities.*" and "*Include the benefits and drawbacks of the reservation system.*". And for NEWTS, the query is a "topic" in the topic model and described in words, phrases or a sentence.

To use LMGQS in the zero-shot setting, it is necessary to convert the queries of diverse formats into natural questions. Without an off-the-shelf tool for this task, we propose to further utilize LMGQS for the query unification task. Specifically, we finetune a BART model to generate queries with the document and summary as input. The finetuned BART model shares the same input and output as the Instruct-GPT used in Section 2.1 to generate queries from generic summarization datasets.

We denote this finetuned model as $G_{d,s \to q}$ and the finetuned model described in Section 3.1 as $G_{d,q \to s}$. Given original query $q$ and document $d$, we first use $q$ as a pseudo "summary" and ask $G_{d,s \to q}$ to produce a query $q'$ of the desired format, i.e., $q' = G_{d,s \to q}(d, q)$. We then use the generated query $q'$ as the input query in the follow-up zero-shot inference to predict summary $s = G_{d,q \to s}(d, q')$.

The query unification is used to generate queries for NEWTS, DUC 2006, and DUC 2007 dataset. We quantitatively and qualitatively verified its effectiveness in section 4.3.

### 3.5 Mutli-document Query Focused Summarization

Since LMGQS contains only single-document QFS data, the fine-tuned model $G_{d,q \to s}$ can generate summaries based on individual document-query pairs. To evaluate zero-shot multi-document QFS, we adopt a straightforward iterative approach from previous works by Baumel et al. (2018); Xu and Lapata (2022). Given a cluster of documents and a query, we first rank documents using term frequency-inverse document frequency, then generate a summary for each ranked document. The final summary is chosen from the top-ranked list. Following the list order, we successively concatenate a summary if its token overlap percentage with the selected summaries is below a threshold, e.g., 50%, until the total length of chosen summaries reaches a predefined token budget (e.g., 250 tokens).

## 4 Evaluation Result

### 4.1 Results on Single-document QFS

The table 4 presents the ROUGE scores and accuracies of stance and relevance for various models on the MultiOpEd dataset. It can be observed that LMGQS BART outperforms other models in both supervised and unsupervised cases, achieving the highest ROUGE scores and stance accuracies in both settings. For relevance accuracy, it also achieves the best in the zero-shot setting and the second best in the supervised setting. This demonstrates the robust performance of LMGQS BART across different settings. Interestingly, in the supervised setting, pre-finetuning on the CNN/DailyMail dataset (CNN/DM BART) actually diminishes performance compared to vanilla BART without pre-finetuning. This result indicates that a general summarization dataset may not always be beneficial for QFS and highlights the necessity for high-quality, large-scale QFS datasets like LMGQS .

Similarly, Table 5 presents the ROUGE scores (R-1, R-2, and R-L) and topic scores on the NEWTS dataset for different models under two categories: Supervised and Zero-shot/Transfer Learning. We used "w/ {query_granularity}" to denote the results using three different granularities for the query: words, phrases, and sentences. For instance, "ProphetNet supervised w/ topic words" refers to the result ProphetNet achieved using a query of topic words. Overall, the LMGQS BART models outperform other baselines in terms of ROUGE scores, with the LMGQS BART w/ topic words model achieving the highest scores in the zero-shot setting and the LMGQS BART w/ topic phrases model obtaining the best results in the supervised setting. Additionally, the LMGQS BART w/ topic sentences model achieves the highest topic score among all models in both zero-shot and supervised settings, closely approaching the topic scores of the ground truth. Without fine-tuning on any supervised data, LMGQS BART exhibits a significant advantage over the supervised ProphetNet models in terms of ROUGE scores and topic scores. The supervised results also reveal that LMGQS remains more beneficial even when some in-domain supervised data (2,400 training samples from NEWTS) is accessible.

Table 6 presents the ROUGE scores on the single-document QFS dataset Debatepedia for various models, classified into unsupervised, supervised, and zero-shot/transfer learning categories.

| Category | Model | R-1 | R-2 | R-L | Topic Score |
|---|---|---|---|---|---|
| Supervised | ProphetNet supervised w/ topic words | 31.9 | 10.8 | 20.7 | 0.136 |
| | ProphetNet supervised w/ topic phrases | 31.6 | 10.4 | 20.2 | 0.147 |
| | ProphetNet supervised w/ topic sentences | 31.4 | 10.0 | 20.0 | 0.163 |
| | CNN/DM BART w/ topic words | 34.5 | 11.5 | 22.1 | 0.176 |
| | CNN/DM BART w/ topic phrases | 34.3 | 11.3 | 22.0 | 0.178 |
| | CNN/DM BART w/ topic sentences | 34.5 | 11.5 | 22.0 | 0.174 |
| | LMGQS BART w/ topic words | 34.4 | 11.9 | 22.5 | 0.178 |
| | LMGQS BART w/ topic phrases | **34.6** | **12.0** | 22.8 | **0.180** |
| | LMGQS BART w/ topic sentences | **34.6** | **12.0** | 22.7 | **0.180** |
| Zero-shot/Transfer Learning | CNN/DM BART | 31.2 | 10.4 | 20.8 | 0.125 |
| | Plug and Play Language Model | 29.6 | 9.1 | 18.8 | 0.148 |
| | Customizable Abstractive Topic-based Summarization | 30.1 | 9.4 | 19.1 | 0.152 |
| | InstructGPT 002 | 32.7 | 10.7 | 21.3 | 0.184 |
| | LMGQS BART w/ topic words | **33.3** | **11.2** | **21.6** | 0.141 |
| | LMGQS BART w/ topic phrases | 32.6 | 10.8 | 21.0 | 0.161 |
| | LMGQS BART w/ topic sentences | 32.4 | 10.5 | 20.9 | **0.187** |
| | Ground Truth | — | — | — | 0.193 |

Table 5: ROUGE scores and topic scores on NEWTS dataset. In NEWTS, the topic query is represented in three formats: word, phrase, and sentence. All baseline results except for InstructGPT 002 are from (Bahrainian et al., 2022). InstructGPT 002 takes topic words as query, which performs the best among all topic formats.

| Category | Model | R-1 | R-2 | R-L |
|---|---|---|---|---|
| Unsupervised | LEAD | 18.1 | 5.6 | 15.9 |
| | LexRank | 17.4 | 5.3 | 15.1 |
| Supervised | DDA | 7.4 | 2.8 | 7.2 |
| | BERTAbs+Rank | 19.2 | 10.6 | 17.9 |
| | BERTAbs+Concat | 26.4 | 11.9 | 25.1 |
| Zero-shot/Transfer Learning | BERTAbs | 13.3 | 2.8 | 2.8 |
| | CNN/DM BART | 21.4 | 6.3 | 18.4 |
| | InstructGPT 002 | 21.8 | 6.5 | 18.8 |
| | GSUM+Query | 21.2 | 6.2 | 18.2 |
| | LaQSUM | 23.5 | 7.2 | 20.6 |
| | LMGQS BART | **23.6** | **7.6** | **21.0** |

Table 6: ROUGE scores on single-document QFS dataset Debatepedia. Baseline results (except for InstructGPT 002) are reported from Xu and Lapata (2022).

LMGQS BART achieves the highest ROUGE scores, surpassing all other models in the zero-shot/transfer learning category.

It is worth mentioning that our model distilled from InstructGPT outperforms the teacher model in the all single-document QFS datasets.

### 4.1.1 Human Study

A recent study by Laskar et al. (2022) discovered that some queries have no relation to the input documents. To investigate this, we conducted a human study comparing the LMGQS BART model's output with the Debatepedia reference. Human annotators were instructed to choose the better summary from two candidates, given the query and the context document. If both summaries were of equal quality or the query was unanswerable from the document, they would mark it as a "Tie." In the blind test, annotators preferred the LMGQS BART model's output 18 times, the reference 15 times, and selected "Tie" 17 times. This indicates that LMGQS has a higher quality compared to existing benchmark datasets like Debatepedia. Additionally, we observe that model finetuned on LMGQS will not just summarize the document but also tends to answer the question by giving a direct answer and giving a stance of supporting or opposing the statement in the query.

### 4.2 Results on Multi-document QFS

Table 7 presents ROUGE scores for various summarization models on multi-document QFS datasets, namely DUC 2006, DUC 2007. The models are categorized into Upper Bound & Baselines, Distantly Supervised, and Few- or Zero-shot Abstractive categories. Note that MARGESUM in this category is a few-shot model while the others are zero-shot ones.

Among the zero-shot abstractive models, LMGQS BART exhibits the second-best performance in terms of ROUGE scores on both DUC 2006 and DUC 2007 benchmarks, only trailing behind its teacher model, InstructGPT 002. We

| Category | Model | DUC 2006 | | | DUC 2007 | | |
|---|---|---|---|---|---|---|---|
| | | R-1 | R-2 | R-SU4 | R-1 | R-2 | R-SU4 |
| Upper Bound & Baselines | Gold | 45.4 | 11.2 | 16.8 | 47.5 | 14.0 | 18.9 |
| | Oracle | 47.5 | 15.8 | 20.2 | 47.6 | 17.1 | 20.9 |
| | Lead | 32.1 | 5.3 | 10.4 | 33.4 | 6.5 | 11.3 |
| | LexRank | 34.2 | 6.4 | 11.4 | 35.8 | 7.7 | 12.7 |
| Distantly Supervised | QuerySum | 41.6 | 9.5 | 15.3 | 43.3 | 11.6 | 16.8 |
| | Bart-CAQ | 38.3 | 7.7 | 12.9 | 40.5 | 9.2 | 14.4 |
| | PQSum | 40.9 | 9.4 | 14.8 | 42.2 | 10.8 | 16.0 |
| Few- or Zero-shot Abstractive | MARGESUM* | 40.2 | 9.7 | 15.1 | 42.5 | 12.0 | 16.9 |
| | QMDSCNN | 31.1 | 6.3 | 10.9 | 34.1 | 7.6 | 12.5 |
| | CNN/DM BART | 38.3 | 7.8 | 13.1 | 40.2 | 9.9 | 14.6 |
| | GSUM+Query | 38.1 | 7.9 | 13.1 | 39.5 | 9.5 | 14.3 |
| | LQSUM | 39.1 | 8.5 | 13.7 | 40.4 | 10.2 | 15.0 |
| | InstructGPT 002 | **41.1** | **9.4** | **15.3** | **42.0** | **10.7** | **16.1** |
| | LMGQS BART | 40.0 | 9.0 | 14.0 | 41.3 | 10.6 | 15.3 |

Table 7: ROUGE scores on multi-document QFS dataset DUC 2006, DUC2007. "*" means the model is few-shot instead of zero-shot. Baseline results (except for InstructGPT 002) are reported from Xu and Lapata (2022).

hypothesize that the primary reason for this is the prevalence of queries in the DUC datasets that are presented in a human-readable instruction format, which inherently favors the instruction-following nature of InstructGPT. Despite being a considerably smaller model, LMGQS BART still demonstrates promising instruction-following capabilities by leveraging our query unification method.

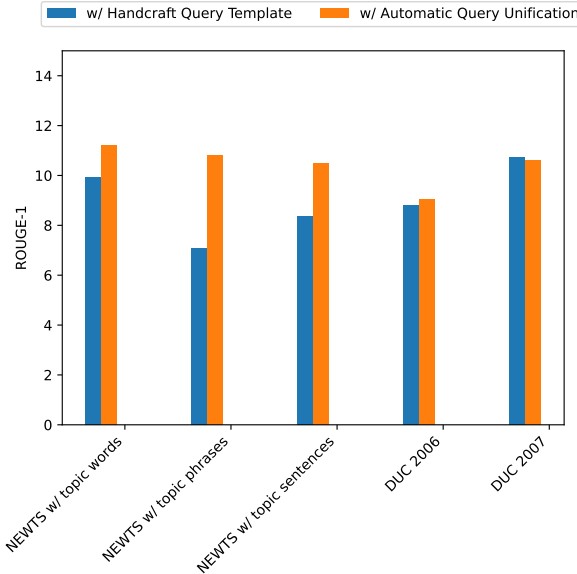

Figure 2: Ablation study on the effect of query unification. For simplicity, we only present ROUGE-2 score in this figure.

### 4.3 Ablation Study of Query Unification

To evaluate the efficacy of query unification, we perform an ablation study to compare the quality of automatically generated queries $q' = G_{d,s \to q}(d, q)$. For comparison, we manually create a query template to transform the query into a natural language question. The template is selected separately for the NEWTS and DUC datasets, and the authors utilize the generation on the development set of these datasets to carefully refine the template, as shown in table 8. In Figure 2, we present a comparison of ROUGE-2 scores between LMGQS BART when employing 1) manually crafted query templates or 2) automatically generated queries from query unification. Other ROUGE scores are presented in table 9 and 10 in appendix. It is evident that query unification holds an advantage over the handcrafted template, despite the latter necessitating access to a validation set and meticulous tuning from human experts.

### 5 Conclusion & Future Works

We introduce a novel large-scale dataset, LMGQS, for query-focused summarization (QFS), addressing the scarcity of large-scale benchmarks in this domain. In order to reduce the laborious human effort required for QFS, we utilize a large-scale language model to generate hidden queries from existing query-agnostic summarization datasets. By performing standard finetuning on LMGQS, we attain state-of-the-art zero-shot performance across multiple QFS benchmarks. The resulting zero-shot

| | NEWTS | DUC |
|---|---|---|
| Template | Template: What are {topical words, phrases, or sentences}? | Replace the instructions such as 'describe', 'indicate', 'include', 'summarize' with 'What are the' |
| Original query | Topics words: winter temperatures, hot weather, a cold morning | Describe the activities of Morris Dees and the Southern Poverty Law Center. |
| Templated query | What are winter temperatures? What are hot weather conditions? What are a cold morning? | What are the activities of Morris Dees and the Southern Poverty Law Center? |

Table 8: The manual template of queries that are not in question format.

model can be further enhanced by finetuning on labeled QFS datasets, achieving the state-of-the-art supervised performance. In addition to QFS, another promising avenue for future research is question answering (QA). Considering the typically extensive context in open domain QA, a query-focused summary of the context could prove advantageous for downstream QA models.

## 6 Ethics Statement

In this study, we acknowledge the ethical concerns related to the use of InstructGPT, a pre-trained language model that has the potential to generate toxic or biased outputs, make up facts, and produce sexual and violent content without explicit prompting (Ouyang et al., 2022). To mitigate these risks, we employed the content filtering function of the OpenAI API calls [3]. This filter runs both the prompt and completion through an ensemble of classification models designed to detect and prevent the output of harmful content. A small fraction of our prepared prompts were filtered by the OpenAI API of InstructGPT, and we discarded the corresponding samples in our proposed LMGQS dataset.

We acknowledge the possibility that the Instruct-GPT model might be discontinued by OpenAI in the future, rendering parts of our research irreproducible. To address this issue, we plan to release the full results returned by the API calls, as well as the prompts for generating the hidden queries. This will enable the research community to construct a higher-quality dataset with more advanced models in the future, reproduce our zero-shot and supervised finetuning results, and further build upon our work.

Regarding the human study, the annotators involved are the paper's two authors who possess proficiency in English and are well-acquainted with the query-focused summarization task. The annotators were tasked with choosing the better candidate summary between two options in a comparison task. To minimize confirmation bias, the order of the candidates was randomized and hidden from the annotators. No payment was involved in this human study, and the authors exercised their best efforts to minimize any inadvertent biases in the annotation process.

## 7 Limitations

One limitation of LMGQS is that it is solely in English, lacking the extensive multilingual coverage of other languages. Additionally, the data creation procedure is cost-inefficient due to the necessity for numerous API calls to InstructGPT. Our human study involves authors of this paper and might be subject to confirmation bias. Lastly, the generated queries are primarily divided into two categories: yes/no queries and wh-queries. A more fine-grained approach to control the types of queries based on the properties of the document and summary is currently lacking. For instance, if a summary emphasizes location over time, a query beginning with "where" would be more appropriate than one starting with "when".

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

# A Appendix

| Model | NEWTS w/ topic words | NEWTS w/ topic phrases | NEWTS w/ topic sentences | DUC 2006 | DUC 2007 |
|---|---|---|---|---|---|
| w/ Handcraft Query Template | 30.13 | 25.84 | 27.94 | **40.54** | 41.14 |
| w/ Automatic Query Unification | **33.27** | **32.64** | **32.42** | 40.02 | **41.32** |

Table 9: Effect of query unification in ROUGE-1

| Model | NEWTS w/ topic words | NEWTS w/ topic phrases | NEWTS w/ topic sentences | DUC 2006 | DUC 2007 |
|---|---|---|---|---|---|
| w/ Handcraft Query Template | 20.25 | 17.23 | 18.79 | **14.38** | **15.49** |
| w/ Automatic Query Unification | **21.58** | **21.01** | **20.85** | 13.99 | 15.32 |

Table 10: Effect of query unification in ROUGE-L (NEWTS dataset) and ROUGE-SU4 (DUC datasets).

| Domain | News | Dialogue |
|---|---|---|
| Document | BOGOTA, Colombia (CNN) – A key rebel commander and fugitive from a U.S. drug trafficking indictment was killed over the weekend in an air attack on a guerrilla encampment, the Colombian military said Monday. Alleged cocaine trafficker and FARC rebel Tomas Medina Caracas in an Interpol photo. Tomas Medina Caracas, known popularly as \"El Negro Acacio,\" was a member of the high command of the Fuerzas Armadas Revolucionarias de Colombia and, according to Colombian and U.S. officials, helped manage the group's extensive cocaine trafficking network. He had been in the cross-hairs of the U.S. Justice Department since 2002. He was charged with conspiracy to import cocaine into the United States and manufacturing and distributing cocaine within Colombia to fund the FARC's 42-year insurgency against the government. U.S. officials alleged Medina Caracas managed the rebel group's sales of cocaine to international drug traffickers, who in turn smuggled it into the United States. He was also indicted in the United States along with two other FARC commanders in November 2002 on charges of conspiring to kidnap two U.S. oil workers from neighboring Venezuela in 1997 and holding one of them for nine months until a $1 million ransom was paid. Officials said the army's Rapid Response Force, backed by elements of the Colombian Air Force, tracked Medina Caracas down at a FARC camp in the jungle in the south of the country. \"After a bombardment, the troops occupied the camp, and they've found 14 dead rebels so far, along with rifles, pistols, communications equipment and ... four GPS systems,\" Defense Minister Juan Manuel Santos said at a news conference. \"The death of 'El Negro Acacio' was confirmed by various sources, including members of FARC itself.\" Medina Caracas commanded FARC's 16th Front in the southern departments of Vichada and Guainia. Established in 1964 as the military wing of the Colombian Communist Party, FARC is Colombia's oldest, largest, most capable and best-equipped Marxist rebel group, according to the U.S. Department of State. E-mail to a friend . Journalist Fernando Ramos contributed to this report. | Emma: I\u2019ve just fallen in love with this advent calendar! Awesome! I wanna one for my kids!\r\nRob: I used to get one every year as a child! Loved them! \r\nEmma: Yeah, i remember! they were filled with chocolates!\r\nLauren: they are different these days! much more sophisticated! Haha!\r\nRob: yeah, they can be fabric/ wooden, shop bought/ homemade, filled with various stuff\r\nEmma: what do you fit inside?\r\nLauren: small toys, Christmas decorations, creative stuff, hair bands & clips, stickers, pencils & rubbers, small puzzles, sweets\r\nEmma: WOW! That\u2019s brill! X\r\nLauren: i add one more very special thing as well- little notes asking my children to do something nice for someone else\r\nRob: i like that! My sister adds notes asking her kids questions about christmas such as What did the 3 wise men bring? etc\r\nLauren: i reckon it prepares them for Christmas \r\nEmma: and makes it more about traditions and being kind to other people\r\nLauren: my children get very excited every time they get one!\r\nEmma: i can see why! :) |
| Summary | 1. Tomas Medina Caracas was a fugitive from a U.S. drug trafficking indictment.\n2. \"El Negro Acacio\" allegedly helped manage extensive cocaine network.\n3. U.S. Justice Department indicted him in 2002.\n4. Colombian military: He was killed in an attack on a guerrilla encampment. | 1. Emma and Rob love the advent calendar.\n2. Lauren fits inside calendar various items, for instance, small toys and Christmas decorations.\n3. Her children are excited whenever they get the calendar. |
| Wh-query | 1. Who was Tomas Medina Caracas?\n2. What was he indicted for?\n3. When was he indicted?\n4. How did he die? | 1. What are Emma and Rob's attitude towards advent calendar?\n2. What does Lauren fit inside advent calendar?\n3. What is the reaction of Lauren's children when they get the calendar? |
| Yes/no Query | 1. Yes: Was Tomas Medina Caracas a fugitive?\n2. No: Did \"El Negro Acacio\" help to fight against drug?\n3. Yes: Was he indicted by U.S. Justice Department?\n4. No: Is he still alive? | 1. Yes: Do Emma and Rob love the advent calendar?\n2. No: Is Lauren unenthusiastic about advent calendar?\n3. Yes: Do Lauren's children enjoy receiving the calendar? |
| Wh-Instruction | For each summary, write a general question about the article that can be answered by it | |
| Yes/no Instruction | For each summary, write a binary question about the article that can be answered by it | |

Table 11: One-shot prompt examples for news and dialogue domain.