# OpenReview forum: "LMGQS: A Large-scale Dataset for Query-focused Summarization"
_EMNLP/2023/Conference — EMNLP 2023 Findings_

### Official Review · Reviewer_udAj · 2023-08-03

**Soundness:** 3

**Excitement:**

3: Ambivalent: It has merits (e.g., it reports state-of-the-art results, the idea is nice), but there are key weaknesses (e.g., it describes incremental work), and it can significantly benefit from another round of revision. However, I won't object to accepting it if my co-reviewers champion it.

**Paper Topic And Main Contributions:**

The paper proposes a novel approach to curate a large dataset for query-focused summarization, Using Large Language Models, queries are constructed and augments to existing summarization datasets - which is then used to train a large model for a generalized QFS task; improvements shown across various QFS datasets.

**Reasons To Accept:**

- Interesting way to augment data


**Reasons To Reject:**

The key novelty seems to be the use of LLMs to augment the queries. However, the quality of the queries thus generated is not validated much - the qualitative evaluation seems to look at the ability of the trained models to match the reference in datasets. I would have loved to see more detailed analysis here.

**Reproducibility:**

3: Could reproduce the results with some difficulty. The settings of parameters are underspecified or subjectively determined; the training/evaluation data are not widely available.

**Reviewer Confidence:**

4: Quite sure. I tried to check the important points carefully. It's unlikely, though conceivable, that I missed something that should affect my ratings.

---

> ### Author Rebuttal · Authors · 2023-08-28
>
> We sincerely thank the reviewer’s feedback! Regarding the quality of the generated queries in LMGQS, we have verified their quality through zero-shot and supervised performance on several query-focused summarization datasets. The state-of-the-art performance of LMGQS confirms the high quality of the queries generated. This is evidenced by the fact that the BART model, fine-tuned on LMGQS, can accurately capture the correspondence between the input query and the relevant summary. Furthermore, the performance gain of LMGQS BART over other strong baselines further confirms its superior quality.

---

### Official Review · Reviewer_Kn1x · 2023-08-05

**Typos Grammar Style And Presentation Improvements:** section 3.4 last paragraph
**Soundness:** 3

**Ethical Concerns:**

Yes

**Excitement:**

4: Strong: This paper deepens the understanding of some phenomenon or lowers the barriers to an existing research direction.

**Justification For Ethical Concerns:**

Given that InstructGPT is a pre-trained model that generates queries based on human-readable instructions, it's unclear whether/how the authors investigated potential biases or controversial language that the model might produce. There could be potential ethical concerns related to biased or inappropriate query generation.

**Paper Topic And Main Contributions:**

This paper presents a novel approach to address the lack of large-scale datasets for query-focused summarization (QFS), a task that involves generating or extracting summaries from documents that directly answer or are relevant to specific queries. The paper proposes a solution to this challenge by leveraging a large-scale pretrained language model, InstructGPT, to generate hidden queries from existing generic summarization datasets. These hidden queries are then combined with document-summary pairs to create a new Query-focused Summarization Dataset. The main contributions of the paper are:

1. Novel Dataset Construction Framework: by utilizing a language model as an annotator to generate hidden queries for existing generic summarization datasets, the authors create a new dataset without the need for labor-intensive human annotation of queries and summaries.
2. LMGQS Dataset.
3. Model Fine-Tuning and Performance: demonstrated the utility of LMGQS by fine-tuning a pretrained language model on it. The model takes the original document concatenated with the generated query as input and is trained to produce the original summary. The resulting model achieves state-of-the-art zero-shot and supervised performance on multiple existing QFS benchmarks.

In summary, this paper addresses the challenge of creating large-scale datasets for query-focused summarization by leveraging a language model to generate hidden queries.

**Questions For The Authors:**

A. Could you elaborate on the rationale behind using the finetuned BART model (Gd,s_q) for query unification, as opposed to alternative methods? How did you determine that this approach would effectively convert queries of diverse formats into natural questions?

B. Have you considered potential challenges or limitations in the unification process? Are there any cases where the generated queries might not align well with the intended natural question format?

C. Given that InstructGPT generates queries based on human-readable instructions, have you investigated potential biases or controversial language that the model might produce? I suggest including a section on potential ethical concerns related to biased or inappropriate query generation.

**Reasons To Accept:**

1. Innovative Dataset Construction. Paper introduces a creative approach to address the lack of large-scale query-focused summarization datasets.

2. Effective Zero-shot and Supervised Performance. The proposed LMGQS BART model achieves sota zero-shot and supervised performance on multiple QFS benchmarks. This outcome underscores the potential of the methodology and dataset in enabling advancements in the development of QFS models.

3. The paper extensively investigated the properties of the LMGQS dataset and conducted an evaluation of the proposed models against a range of baseline models. Human study also involved.

4. Addressing Query Unification. The approach to query unification addresses a critical challenge in zero-shot QFS across diverse query formats. While there may be room for deeper analysis, the paper lays the foundation for a novel solution and demonstrates its superiority over handcrafted templates.

**Reasons To Reject:**

1. The query unification approach for adapting diverse query formats to natural questions relies on finetuning a BART model. However, the paper does not thoroughly address the challenge of ensuring accurate and meaningful query conversions across different datasets. Ambiguities in query transformation may lead to misaligned summaries, impacting the overall quality of the model's outputs.
2. There could be biases in the generated queries from InstructGPT.
3. Insufficient discussion on the sensitivity of hyperparameters on final performance.


**Reproducibility:**

4: Could mostly reproduce the results, but there may be some variation because of sample variance or minor variations in their interpretation of the protocol or method.

**Reviewer Confidence:**

3: Pretty sure, but there's a chance I missed something. Although I have a good feel for this area in general, I did not carefully check the paper's details, e.g., the math, experimental design, or novelty.

---

> ### Author Rebuttal · Authors · 2023-08-28
>
> We sincerely thank the reviewer’s feedback! We will follow the review’s suggestions on the typo and presentation issues.
>
> **Question A** about the rationale behind using the finetuned BART model: We first addressed the query unification with human-designed templates (section 4.3). However, this method requires humans to read the raw queries and cannot be easily extended to unseen datasets. To automatically generate queries of a unified format, we utilize finetuned BART model (Gd,s_q) and input the raw query as "pseudo" summary. We manually checked the queries generated and most of them are of excellent quality.
>
> **Question B** about the potential challenges or limitations in the unification process: The potential challenge is that there is no way to directly assess the quality of the generated queries. We did observe a small number of generated queries that did not align well with the intended natural question format. However, most of them are of decent quality. And the end-to-end performance on the generated summaries could prove the effectiveness of our query uniform method, especially compared with the carefully handcrafted templated queries. (section 4.3)
>
> **Question C** about ethical concerns: We address the ethical concerns in the "Ethics Meta Review" of this page. And we will follow the reviewer's advice to include the discussion there in our revision.

---

### Official Review · Reviewer_HLfb · 2023-08-12

**Soundness:** 3

**Excitement:**

3: Ambivalent: It has merits (e.g., it reports state-of-the-art results, the idea is nice), but there are key weaknesses (e.g., it describes incremental work), and it can significantly benefit from another round of revision. However, I won't object to accepting it if my co-reviewers champion it.

**Missing References:**

The paper [Data Augmentation for Abstractive Query-Focused
Multi-Document Summarization](https://ojs.aaai.org/index.php/AAAI/article/view/17611/17418) suggest data augmentation for MD-QFS and two automatically derived QFS datasets, _QMDSIR_ and _QMDSCNN_. It is desired to compare _LMGQS_ to them.

**Paper Topic And Main Contributions:**

The paper presents a protocol to automatically derive a QFS dataset from a generic summarization dataset by utilizing generative models to generate questions for every summary sentence together with a new large QFS dataset (named _LMGQS_), by applying the protocol to 4 known summarization datasets.
The authors further fine-tune BART using _LMGQS_ and compare the effectiveness of the new dataset for training QFS models in the ZS and supervised settings.
The paper discusses the problem of unifying queries between different QFS datasets.

**Questions For The Authors:**

-

**Reasons To Accept:**

* A new large scale dataset for QFS with more than 1.1M samples and a protocol for deriving it
* Presenting the query unification task and providing a method for handling it

**Reasons To Reject:**

1. It is desired to have more evidence or analysis supporting the training effectiveness property of the dataset or other key properties
that will explain the importance and possible use-cases of _LMGQS_ over other QFS datasets.

2. Several unclear methods affecting readability and reproducibility:
* "To use LMGQS in the zero-shot setting, it is necessary to convert the queries of diverse formats into natural questions." (L350) please explain why.
* "Specifically, we finetune a BART model to generate queries with the document and summary as input." (L354) - how did you FT? what is the training set and any relevant hyper-parameters for reproducing the results.
* "we manually create a query template to transform the query into a natural language question." (L493) - what are the templates? what are the query template and several examples.

**Reproducibility:**

4: Could mostly reproduce the results, but there may be some variation because of sample variance or minor variations in their interpretation of the protocol or method.

**Reviewer Confidence:**

4: Quite sure. I tried to check the important points carefully. It's unlikely, though conceivable, that I missed something that should affect my ratings.

**Typos Grammar Style And Presentation Improvements:**

* Figure 2 - hard to understand the exact ROUGE numbers, better have number labels on the bars or inline the results on the relevant section text.

---

> ### Author Rebuttal · Authors · 2023-08-28
>
> We sincerely thank the reviewer’s feedback! We will follow the review’s suggestions on the typo and presentation issues.
>
> To address the **missing references of QMDSIR and QMDSCNN**: Authors of QMDSIR and QMDSCNN also tested the transferability of their datasets on DUC 2006 and 2007. The ROUGE-2 of their best performing model is **6.28** and **7.60** for DUC 2016 and DUC 2017 respectively. The ROUGE-2 for our LMGQS BART is **9.0** and **10.6**, significantly outperforming the model trained on QMDSIR and QMDSCNN. We will add the reference and empirical comparison to our revision.
>
> To answer the "Several unclear methods affecting readability and reproducibility":
>
> **About the necessity to convert queries into a unified format**
>
> The LMGQS dataset is designed to work with queries in the form of natural language questions. (see table 10 for query examples) However, different QFS datasets have different query formats. (table 3) For example, Debatepedia has the query format of a natural question, which is the same as LMGQS, while the majority of queries in DUC datasets are instructions such as “Discuss conditions on American Indian reservations or among Native American communities.” and “Include the benefits and drawbacks of the reservation system.” To use LMGQS in the zero-shot setting, it is necessary to convert these diverse query formats into natural language questions so that they can be processed by the LMGQS model. This process is called query unification.
>
> **About the finetuning in query unification**
>
> We propose to further utilize LMGQS for the query unification task by fine-tuning a BART model to generate queries with the document and summary as input. Basically, this is what Instruct-GPT was prompted to do in Section 2.1 of the paper. We input query of diverse format as pseudo "summary" and the generated query will be in a unified format. This approach allows for automatic conversion of diverse query formats into natural language questions that can be processed by the LMGQS model. The hyper-parameters for finetuning are the same as our main LMGQS BART model, which is described in section 3.1. We will make it clear in our revision.
>
> **About query template and examples**
>
> We show them below for NEWTS and DUC datasets. And we will add a table showing this in our revision.
> - NEWTS
>   - Template: What are {topical words, phrases, or sentences}?
>   - Original query: winter temperatures, hot weather conditions, a cold morning
>   - Templated query: What are winter temperatures? What are hot weather conditions? What are a cold morning?
> - DUC:
>   - Template: Replace the instructions such as 'describe', 'indicate', 'include', 'summarize' with 'What are the'
>   - Original query: Describe the activities of Morris Dees and the Southern Poverty Law Center.
>   - Templated query: What are the activities of Morris Dees and the Southern Poverty Law Center?

---

### Meta-Review · Area_Chair_vMj2 · 2023-09-25

**Recommendation:** 2

**Metareview:**

Considering ethics issues pointed by  Ethics Chairs, I would suggest rejection of this paper.

---

### Meta-Review · Senior_Area_Chairs · 2023-09-30

**Recommendation:** 3

**Metareview:**

Ethical concerns seemed to have been resolved by authors and ethics committee (only acknowledgement was needed). Reviewers seem to agree that this work's proposed methods are novel and have merits. Some concerns were still flagged but overall they were positive.

---

### Decision · Program_Chairs · 2023-10-07

**Decision:**

Accept-Findings

**Comment:**

Considering ethics issues pointed by  Ethics Chairs, I would suggest rejection of this paper.|Ethical concerns seemed to have been resolved by authors and ethics committee (only acknowledgement was needed). Reviewers seem to agree that this work's proposed methods are novel and have merits. Some concerns were still flagged but overall they were positive.